# Bioinformatic Analysis of Recurrent Genomic Alterations and Corresponding Pathway Alterations in Ewing Sarcoma

**DOI:** 10.3390/jpm13101499

**Published:** 2023-10-15

**Authors:** Adam Rock, An Uche, Janet Yoon, Mark Agulnik, Warren Chow, Sherri Millis

**Affiliations:** 1City of Hope Comprehensive Cancer Center, 1500 E. Duarte Rd., Duarte, CA 91010, USA; jyoon@coh.org (J.Y.); magulnik@coh.org (M.A.); 2Alameda Health System, 1411 E. 31st St., Oakland, CA 94602, USA; anuche@alamedahealthsystem.org; 3UCI Health, 101 The City Drive, South Orange, CA 92868, USA; wachow@uci.edu; 4Foundation Medicine, Inc., 150 Second St., Cambridge, MA 02141, USA; sherri.millis@gmail.com

**Keywords:** ewing sarcoma, genomic alterations, fibroblast growth factor receptor, genomics, comprehensive genomic profiling

## Abstract

Ewing Sarcoma (ES) is an aggressive, mesenchymal malignancy associated with a poor prognosis in the recurrent or metastatic setting with an estimated overall survival (OS) of <30% at 5 years. ES is characterized by a balanced, reciprocal chromosomal translocation involving the *EWSR1* RNA-binding protein and *ETS* transcription factor gene (*EWS-FLI* being the most common). Interestingly, murine ES models have failed to produce tumors phenotypically representative of ES. Genomic alterations (GA) in ES are infrequent and may work synergistically with *EWS-ETS* translocations to promote oncogenesis. Aberrations in fibroblast growth factor receptor (*FGFR4*), a receptor tyrosine kinase (RTK) have been shown to contribute to carcinogenesis. Mouse embryonic fibroblasts (MEFs) derived from knock-in strain of homologous *Fgfr4G385R* mice display a transformed phenotype with enhanced TGF-induced mammary carcinogenesis. The association between the *FGFRG388R* SNV in high-grade soft tissue sarcomas has previously been demonstrated conferring a statistically significant association with poorer OS. How the *FGFR4G388R* SNV specifically relates to ES has not previously been delineated. To further define the genomic landscape and corresponding pathway alterations in ES, comprehensive genomic profiling (CGP) was performed on the tumors of 189 ES patients. The *FGFR4G388R* SNV was identified in a significant proportion of the evaluable cases (*n* = 97, 51%). In line with previous analyses, *TP53* (*n* = 36, 19%), *CDK2NA/B* (*n* = 33, 17%), and *STAG2* (*n* = 22, 11.6%) represented the most frequent alterations in our cohort. Co-occurrence of *CDK2NA* and *STAG2* alterations was observed (*n* = 5, 3%). Notably, we identified a higher proportion of *TP53* mutations than previously observed. The most frequent pathway alterations affected *MAPK* (*n* = 89, 24% of pathological samples), *HRR* (*n* = 75, 25%), *Notch1* (*n* = 69, 23%), Histone/Chromatin remodeling (*n* = 57, 24%), and *PI3K* (*n* = 64, 20%). These findings help to further elucidate the genomic landscape of ES with a novel investigation of the *FGFR4G388R* SNV revealing frequent aberration.

## 1. Introduction

Ewing Sarcoma (ES) is the second most common bone cancer in children and young adults with approximately 1.5 cases per million [1,2]. Multimodality therapy incorporating local and systemic treatments has drastically improved 5-year survival of patients with local disease to more than 70% [3]. Despite advances in available treatments, recurrent or metastatic ES carries a poor prognosis with 5-year overall survival estimated at <30% [4,5,6]. ES is characterized by a balanced reciprocal chromosomal translocation t(11;22) (q24;12) between the Ewing sarcoma RNA-binding protein 1 (*EWS*) gene, *EWSR1*, and members of the E26 transformation-specific (*ETS*) gene family. This results in an in-frame fusion of the *EWS* gene to *ETS* to generate a hybrid fusion gene, *EWS-FLI1* [7]. While Friend Leukemia Insertion (*FLI1)* gene is the most common involved gene, other members of the *ETS* gene family include *ERG, ETV1, E1AF*, and *FEV* [7,8,9,10,11,12]. Despite knowledge of this oncogenic chimeric transcript, there remains a paucity of data surrounding collaborating genetic alterations that may impact sarcoma development or clinical outcomes.

Reproducing the EWS-ETS chimeric transcript in genetically engineered mouse models (GEMM) has presented several challenges. In a comprehensive analysis of transgenic and non-transgenic mouse models, researchers were unable to reproduce ES, instead resulting lack of phenotypic expression or embryonic death [13]. Only immortalized fibroblasts engineered to express *EWS-FL1* have successfully formed tumors morphologically resembling ES [14]. More recently, MSCs transfected with ribonucleic protein (RNP/Cas9 complexes) led to readily detectable *EWSR1-FLI1* translocation positive cells. Of note, viable clones were observed only after co-occurring alteration of *CDKN2A.* Furthermore, additional chromosomal translocations reminiscent of chromoplexy, a loop-like rearrangement previously noted in ES tumors, were observed [15]. These data support the notion that additional genomic alterations (GAs) may aid in ES transformation. 

Additional GAs and associated pathway alterations have been evaluated for their potential to induce sarcoma oncogenesis [16,17,18]. Previously identified and recurrent GAs including at the *STAG2* gene, which encodes a subunit of the cohesion complex responsible for regulation of sister chromatid separation undergoing cell division [17,18,19,20]. Recent genomic analysis of pediatric ES samples identified several recurrent GAs including *STAG2* alterations [18]. Both *STAG2* variant and loss of expression were noted in high proportion (36% and 70%, respectively) [18]. In murine models with mesenchymal stem cells encompassing inducible *EWS-FLI1* transgenes, *STAG2* inhibition has been associated with sarcoma formation and reduced survival [21]. Recurrent, somatic mutations in *TP53* and *CDK2NA* have also been observed [17,18,19,20,21,22]. *CDK2NA* and *TP53* have both been postulated to mediate cell cycle regulation in conjunction with *EWS-ETS* [23]. In previous analyses of ES genomics, a statistically significant co-association between *TP53* and *CDK2NA* and this association was characterized by a poorer prognosis [20]. Interestingly, transfected MSCs expressing *EWSR1, FLI1*, and *TP53* failed to result in significant colony growth. Alternatively, co-alteration of *CDK2NA* or *STAG2* resulted in a significant increase in size furthering intrigue into how additional, recurrent variants may impact ES development [15]. Furthermore, CGP has utility in evaluating tumor mutational burden (TMB) and microsatellite instability (MSI), which are both clinically relevant in predicting response to checkpoint inhibitors. However, sequencing analyses of ES have been characterized by a lack of TMB and no therapeutic strategies are available to date [16,17,18].

Fibroblast factor receptor 4 (*FGFR4*) has been investigated and therapeutically targeted in a variety of solid tumors. Acting as a receptor tyrosine kinase (RTK) protein, *FGFR4* is activated by a family of ligands, fibroblast growth factors (FGFs), at the extracellular domain resulting in intracellular transmission signals via transmembrane domain and intracellular tyrosine kinase [24]. The *FGFR4Gly38Arg* (G388R) single nucleotide variant (SNV) results in the substitution of arginine (Arg) for glycine (Gly) in the transmembrane domain of the receptor. The reported prevalence of the *FGFR4G388R* SNV is approximately 32% in the general population [25]. This SNV has been found to significantly increase the risk of breast and prostate cancer with a capacity to increase motility in mammary tumor cells and has been postulated to increase the risk of cancer and promote metastasis [26,27,28,29,30]. MEFs derived from knock-in strain of homologous *Fgfr4G385R* mice display a transformed phenotype with increased *STAT3* signaling confirmed in vivo [29,30]. Additionally, the *FGFR4G388R* SNV has been associated with *FGFR4* protein damage and increased *FGFR4* expression [25,28]. The association between the *FGFRG388R* SNV in high-grade soft tissue sarcomas has previously been shown to have a deleterious effect on overall survival [31]. However, the significance of this has yet to be evaluated in ES. In this study, we evaluated the frequency of *FGFR4G338R* SNV, other recurrent GAs, and their corresponding pathway alterations in ES with CGP.

## 2. Materials and Methods

As part of routine clinical care, formalin-fixed and paraffin-embedded (FFPE) tissue from 189 Ewing sarcoma patients were sent to Foundation Medicine for CGP between 2012 and 2018 (Foundation Medicine, Cambridge, MA, USA). The presence of the *EWS-FLI-1* fusion gene was confirmed during Foundation Medicine testing. The cohort was comprised of only patients with confirmed *EWS-FL1* fusion gene. FoundationOne^®^ Heme CGP evaluated GAs including base substitutions, insertions and deletions (indels), gene amplifications, copy number alterations (CNAs), gene fusions, rearrangements (REs), and single nucleotide variations (SNVs) by next generation sequencing (NGS). 189 samples were assayed by hybrid-capture based CGP, including 406 DNA-sequenced genes in addition to 265 RNA-sequenced genes commonly reported to be rearranged in cancer, which was previously documented by He et al. [32]. At least 50 ng of DNA were analyzed by next generation sequencing (NGS) via Illumina HiSeq. Characterized by mutations/Mb, TMB was assessed using a minimum 1.4 Mb sequenced DNA. An algorithm evaluating 95 loci was used to ascertain MSI status. Information regarding the clinical context, including stage and treatment, were not typically submitted with the specimen; therefore, the clinical status, outcomes, and source acquisition (primary tumor, metastasis, or recurrence) information was largely unknown to Foundation Medicine. All GAs were included in the final analysis after excluding variants of unknown significance (VUS). Approval for this study was procured from the Western Institutional Review Board (Protocol No. 20152817) including a waiver of informed consent in addition to a HIPAA waiver of authorization.

## 3. Results

Tissue from 189 ES clinical samples were analyzed with CGP. Demographic information is described in Table 1. Our patient population was comprised of 113 (60%) male and 76 (40%) female patients. Median age of patients included in the study was 20 years (range, 0 to 70 years). The number of pediatric and adolescent young adult patients were 75 (40%) and 87 (46%), respectively, representing a majority of the patients analyzed. Adults comprised 27 (14%) of the total clinical samples.

CGP identified several GAs in ES represented by the heat map in Figure 1. Genes that were altered in at least three patients are illustrated. Variants of unknown significance (VUS) were excluded from the final analysis. On average, there were 7 GAs per case. All included cases were characterized by the *EWS-FL1* translocation.

The highest incidence of pathway alterations affected MAPK (*n* = 89, affecting 24% of individual samples), *HRR* (*n* = 75, 25%), *Notch1* (*n* = 69, 23%) Histone/Chromatin remodeling (*n*= 57, 24%), and *PIK3* (*n* = 64, 20%) with additional pathways illustrated in Table 2. These percentages represented the proportion of samples affected by a GA in particular molecular pathway with many samples demonstrating several GAs in the same pathway. However, recurrent genomic variants in each pathway were infrequent. Alternatively, individual genes that were noted to be altered in high proportions included *TP53* (*n* = 36; 19%), *CDKN2A/B* (*n* = 33, 17%), and *STAG2* (*n* = 23, 12%) as illustrated in Figure 2. The EWSR1-ETS translocation was observed in 100% of evaluated samples. Additional GAs noted in high proportion affecting *PCLO*, *RAD21*, and *KDMSC* as demonstrated in Figure 2. CNVs noted involved chromosome 1q (*n* = 5, 2.6%) and chromosome 8q (*n* = 15, 7.9%). No CNVs were observed involving chromosome 16q.

The *FGFR4G388R* SNV was found in over half of the evaluated samples (*n* = 97, 51%) and coincided with GAs in high frequency (Table 2). Pathways commonly noted to be altered in the presence of the *FGFR4G388R* SNV were *MAPK* (*n* = 63, 33%), *Notch1* (*n* = 37, 20%), *HRR* (*n* = 36, 19%), Histone/chromatin remodeling (*n* = 33, 18%), and Cyclin pathways (*n* = 28, 15%). Additional recurrent GAs noted in combination with the *FGFR4G388R* SNV are displayed in Table 2. Of the ES samples analyzed, 0% of the evaluated samples were characterized by high TMB or microsatellite instability (MSI).

Observed GAs included single nucleotide variations (SNV), copy number (CN) alterations, and rearrangements (RE). Overall, SNVs were noted in the largest proportion accounting for 81% of observed GAs, with a smaller proportion resulting from CNs (18%) and REs (4%). GAs affecting the MAPK pathway were comprised of SNVs, CNs, and REs at 80%, 18%, and 2% respectively. Similarly, GAs impacting HRR included 78% SNVs and 22% CNs, without any observed REs. Both Notch1 and SWI/SNF pathway GAs were exclusively comprised of SNVs (100%). *WNT* represented the only pathway most frequently characterized by CN alterations (80%). Although recurrent, pathogenic variants were observed in potential tumor-agnostic targets (*NTRK, RET),* none of the observed variants were rearrangements/fusions for which the current FDA approvals exist.

## 4. Discussion

As reviewed by Chae et al., small molecule inhibitors and monoclonal antibodies targeting various *FGFR*s are currently under investigation in multiple solid tumor types [33]. In a phase 1 trial of Fisogatinib, a type 1 irreversible inhibitor of *FGFR4,* an overall response rate (ORR) of 17% and median duration of response (DOR) of 5.3 months were achieved in patients with hepatocellular carcinoma [34]. The *FGFR4G388R* SNP has been evaluated in high-grade soft tissue sarcoma and associated with a poor prognosis [31]. These findings suggest a potential for enhanced oncogenesis in the presence of *FGFR4G388R* SNV. Recently, an evaluation of *FGFR* alteration targeting is underway advanced sarcomas harboring pre-specified alterations in *FGFR1-4* (ClinicalTrials.gov Identifier:NCT04595747). In our analysis, the *FGFR4G388R* SNV was detected in over half of the samples evaluated in our analysis. This is notably higher than would be expected in the general population based on previous large-scale analysis [25]. Owing to the lack of matched germline mutational testing, it is difficult to ascertain the origin of the SNP in our particular cohort. Secondary GAs were identified in more than one third of patients included in our study (*n* = 92, 37%). The *FGFR4G388R* SNV often co-occurred with recurrent GAs. How the *FGFR4G388R* SNV may affect GAs and pathways implicated in sarcoma formation is not clearly understood. Interestingly, while the FGFR4G388R SNV co-occurrence appeared random with most pathway alterations, the *PI3KCA* pathway was disproportionately altered in the absence of *FGFR4G388R* SNV co-alteration. This could suggest that downstream signaling alterations may modulate ES formation in patients without an existing *FGFR4* alteration. Further investigation with clinical correlative data is necessary to better understand the potential pathogenicity of this particular SNV in ES.

In line with previous analyses, *TP53* (*n*= 36, 19%), *CDK2NA/B* (*n* = 33, 17%), and *STAG2* (*n* = 22, 11.6%) represented the most frequently altered genes in our cohort after excluding VUS. Our data illustrated a higher proportion of *TP53* mutations (19%) when compared to previous analyses of the genomic landscape in ES, which have demonstrated *TP53* mutations in approximately 5.7 to 7% of tumor samples [17,20,22]. Potentially, this could be a result of a larger sample size, older median age when compared to previous analyses, or related to selection bias associated with tertiary referral with previous treatment. *TP53* is a tumor suppressor gene and its loss of function is frequently implicated in tumor development. *TP53* mutational loss has previously been identified in ES and associated with higher TMB and shorter overall survival (OS) [35].

Additional genes frequently mutated in our cohort included *CDK2NA/B* and *STAG2*. Interestingly, in previous genomic analyses of ES, a mutual exclusivity appears to exist between these two alterations [20]. In our analysis, *CDK2NA* and *STAG2* were mutated in 17% and 12% of cases, respectively. Co-occurrence of *CDK2NA* and *STAG2* variants appeared in 5 (3%) of the cases. In a recent evaluation of *STAG2* in ES, researchers concluded that STAG2 loss affects the gene-regulatory architecture resulting in promotion of disease progression [36]. Furthermore, *STAG2* is thought to function through its interaction with *CTCF* subsequently impacting gene expression regulated by EWS-FLI1 [37]. Interestingly, *CTCF* has also been shown to interact with *CDKN2A* locus, regulating transcription [38].

GAs affecting the following pathways were most prevalent in our analysis: MAPK, HRR, Notch1, Histone/Chromatin remodeling, and *PI3K* with additional pathways illustrated in both Table 2 and Figure 1. Affected genes identified by CGP included potential oncogenes and tumor suppressor genes. *PI3KCA* and *MAPK* pathway alterations represented two of the most common pathways altered in our evaluation. *PI3K* signaling pathway is frequently implicated in oncogenesis and typically mediated through loss of the inhibitory protein, *PTEN* [39]. Previous investigations have demonstrated dysfunctional growth factor signaling in ES cells with *PI3K* activity enhanced by *PIK3R3* and loss of *PTEN* [40]. Furthermore, PTEN status was linked with variable response to microtubule inhibition. MEK/MAPK pathway was analyzed in ES cells with disruption of *MEK/MAPK* or *PI3K* pathways via insulin growth factor-1 receptor (IGF-1R) neutralizing antibodies being associated with functional consequences including delayed time to primary tumor development and attenuated growth [41]. In vitro analysis has demonstrated enhancement of Actinomycin-D-induced apoptosis with combined administration of *PI3K* and *MAPK* inhibitors resulting in suppression of tumor growth [42]. Considering the evolving role for therapeutic *PI3K* inhibition, further investigation is warranted and currently underway (ClinicalTrials.gov identifier: NCT05440786, NCT04129151). Perhaps, CGP would play a role in more appropriate patient selection for these agents.

*Notch* signaling is highly conserved through evolution in multicellular organisms resulting in control of cellular proliferation, differentiation, and apoptosis. The *Notch* pathway can influence development of neighboring cells via juxtracrine signaling. Four receptors (*Notch*1-4) act in a canonical receptor-ligand interaction resulting in a series of cleavages to the Notch receptor leading to release of the *Notch* intracellular domain (NICD) [43]. Thereafter, NICDs translocate into the nucleus interacting with *CBF-1/Su(H)/LAG1* (*CSL*) transcription factors that together recruit additional transcriptional co-activators (Co-A) and displacement of transcription co-repressors (Co-Rs) [44]. Furthermore, Notch signaling impacts tumor vasculature and immune infiltration in the tumor microenvironment [44]. Inhibition of Notch signaling has been a developing focus of cancer research. LY3039748, an oral Notch inhibitor, acts by preventing release of the NICD and thereby decreasing downstream signaling and subsequent biologic effects [45]. In our analysis, the Notch1 pathway was frequently altered, often co-occurring with the *FGFR4G388R* SNP as demonstrated in Table 2.

*HRR* is pivotal in repair of double-strand breaks generated during crosslinking of DNA. Deficiency of *HRR* results from both germline alterations in *BRCA1* and *BRCA2*, as well as with genetic or epigenetic inactivation with somatic variants contributing to a BRCA-like phenotypic expression [46]. Synthetic lethality, in which cancer cells deficient in *HRR* have unrepaired DNA break due to inhibition, has been successfully exploited with the use of Poly (ADP-ribose) polymerase (PARP) inhibitors [47]. ES cells have previously been shown to increase R-loop accumulation that are associated with homologous recombination [48]. Expression of *EWS-FL1* or *EWS-ERG* lead to significant reduction in homologous recombination activity thought to be secondary to loss of *EWSR1* function [48]. Furthermore, functional *BRCA1* deficiency was noted in ES cells suggesting a possible role for therapeutic strategies involving PARP-1 inhibition. ES mouse xenografts have been shown to be highly sensitive to PARP-1 inhibition with *EWS-FL1* transcription mediated through PARP1 [49]. Furthermore, PARP inhibition has been reported to potentiate the effects of cytotoxic chemotherapy, including temozolomide and topoisomerase-1 inhibitors that induce base excision repair [50]. In a multicenter, phase 1 study evaluating niraparib, a PARP1 inhibitor, in combination with either temozolomide (arm 1) or irinotecan (arm 2), patients in arm 1 achieved a median PFS of 9 weeks and those treated with irinotecan achieved a PFS of 16.3 weeks with ORR 8.33 [51]. Our cohort identified multiple pathogenic genomic alterations impacting *HRR* including *CHEK2* (*n*= 6, 3%), *BRCA1* (*n* = 5, 3%), *BRCA2* (*n* = 5, 3%), *BARD1* (*n* = 1, 1%), *CHK1* (*n* = 1, 1%), and *RAD51D* (*n* = 1, 1%). Potentially, CGP may help to identify a subpopulation of patients with enhanced susceptibility to PARP inhibition. Multiple clinical trials are actively investigating the role of targeting homologous recombination deficiency (HRD) in ES via *CHK1* (ClinicalTrials.gov identifier: NCT05275426) and PARP inhibition (ClinicalTrials.gov identifier: NCT01858168, NCT04901702).

GAs of *MSH3* and *RUNX1T1* in a small proportion of samples represented an additional subgroup of the *DDR* pathway in our analysis. *MSH3* functions as a heterodimer with *MSH2* and is utilized in mismatch repair of detected insertion-deletion loops. Somatic alterations in *MSH3* are associated with dysfunctional MMR and microsatellite instability [52]. *RUNX1T1* functions as a transcriptional co-repressor and interacting with histone deacetylases (*HDAC*s) and is involved in multiple cellular processes including neuronal differentiation, microglial proliferation, endothelial angiogenesis, and adipocyte differentiation [53,54,55,56]. Further investigation is needed to evaluate the impact of DNA mismatch repair and *DDR* pathway as they relate to the pathogenesis of ES.

Additional pathways frequently altered in our analysis included switch/sucrose-nonfermentable (*SWI/SNF*), epigenetic modification, and the cyclin pathway. The *SWI/SNF* chromatin remodeling complex, a highly conserved ATP-dependent chromatin remodeling complex influencing transcriptional activity, is identifiable by next generation sequencing in genes including *ARID1A, EZH2*, *INI1/SMARCB4*, *SMARCA4* among others [57,58]. Murine models of *ARID1A*-deficient cells lead to reduction of *SWI/SNF* regulation of enhancers associated with tumor generation [59]. *ARID1A* variants were present in 4% of ES samples analyzed in this study. The role of *SWI/SNF* complex in oncogenesis is an active area of interest highlighted with the development of tazemetostat, a selective inhibitor of *EZH2*. In an open-label, phase I trial investigated tazemotostat in relapsed or advanced solid tumors, tazemotostat was found to be well-tolerated with promising activity, notably in epithelioid sarcoma patients [60,61]. Chromatin remodeling resulting from histone modification has been implicated in the disruption of transcriptional regulation thereby contributing to carcinogenesis [62]. *LSD1*, an epigenetic modifying demethylase, has previously been shown to be upregulated in ES [63]. *LSD1* is thought to contribute to ES formation and overall survival. In a phase 1, non-randomized trial, SP-2577 (Seclidemstat), a reversible *LSD1* inhibitor, is being evaluated in treatment of recurrent or refractory ES (NCT03600649). In previous evaluation of epigenetic modification, functional genomics revealed an activated *cyclinD1/CDK4* pathway with potential sensitivity to chemical inhibition [64]. The cyclin pathway functions through upregulated *cyclin D1* binding to cyclin dependent kinases, such as *CDK4* and *CDK6*. Subsequent phosphorylation of RB, a cell cycle regulator, mediates cell cycle progression after dissociating from G1 to S phase-promoting transcription factors [65,66]. Each of these pathways were noted to be altered frequently in our cohort. How they impact tumor formation and or disease progression remains to be elucidated.

The utility of immunotherapy in Ewing Sarcoma is unclear and remains investigational. ES does not appear to be rich in tumor-infiltrating lymphocytes (TIL), nor does it exert high levels of PD-L1 expression. Interestingly, ES cells have been observed to have a high frequency of partial or complete absence of HLA class I expression, which has been associated with absence of CD8+ T cell infiltration [67]. Tumor mutational burden has been shown to be among the lowest observed in all tumor types [17,18]. Trials involving monoclonal antibodies directed against PD-1 or PD-L1 have shown limited activity in patients with ES, which may be attributable to an overall low tumor mutational burden or PD-L1 expression in ES cells [17,18,20,68]. Furthermore, previous studies of microsatellite instability (MSI) in Ewing Sarcoma have demonstrated low prevalence [69,70,71]. Similarly, our analysis recapitulates previous findings of low tumor mutational burden and low PD-L1 expression [17,18,20,68].

## 5. Conclusions

In summary, alterations of the *FGFR4G388R* SNV were demonstrated alone and in conjunction with additional GAs in a high proportion of a large cohort of ES tumors. *TP53* was mutated in higher proportion than previously reported. Additional recurrent GAs included *STAG2* and *CDKN2A* with demonstrated co-occurrence in a small proportion of the evaluable cases. The role and interplay between these genomic alterations are unclear and warrant further investigation. Unfortunately, patient clinical outcomes were not available for this cohort to further define the prognostic or predictive implications, which represents a major limitation of the analysis. Furthermore, age-matched controls were not available to analyze for a true enrichment of the *FGFR4G388R* SNV in this population. The lack of validation studies analyzing patient-derived cell lines represents another major limitation. Variant allele frequency (VAF) was not evaluable and may affect the pathogenicity in a continuous manner rather than purely in a binary manner warranting further exploration. Future investigation should be directed at the association of these GAs and their potential impact on clinical correlates including grade, time to progression, frequency of metastasis, and treatment response.

## Figures and Tables

**Figure 1 jpm-13-01499-f001:**
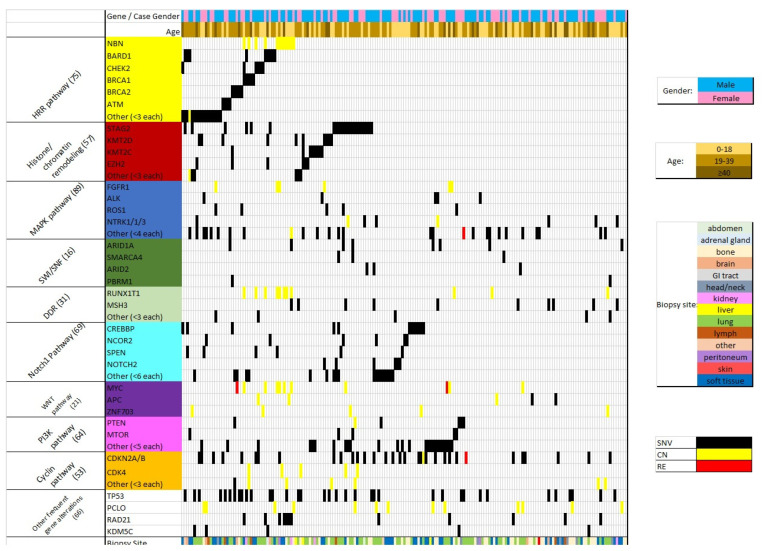
GAs in ES. The heat map represents proportion of GAs noted in the test population categorized by gender, age group, and site of biopsy. Pathogenic single nucleotide variations (SNV) are displayed in black, copy number alterations (CN) in yellow, and rearragements (RE) in red.

**Figure 2 jpm-13-01499-f002:**
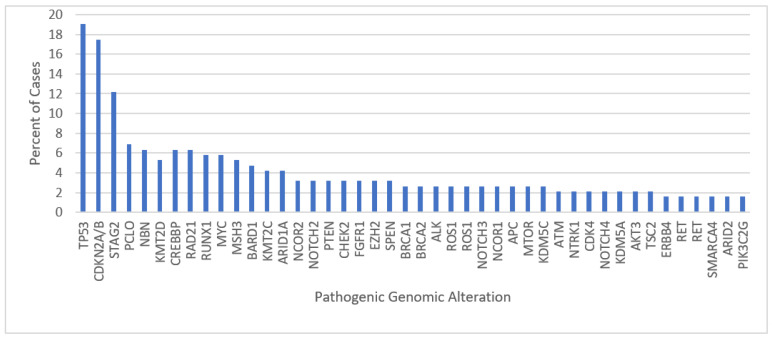
Long tail plot of all recurrent genomic alterations (GA) observed in at least 2 cases as a percentage of the total cases impacted.

**Table 1 jpm-13-01499-t001:** Demographic information including gender and age of included samples.

	Total	Percent
**Gender**		
Male	113	60%
Female	76	40%
**Age**		
0–18	75	40%
19–39	87	46%
≥40	27	14%

**Table 2 jpm-13-01499-t002:** Recurrent variants and their relation with FGFRG388R variant.

Genomic Alteration	No FGFR4 G388R Variant	FGFR4 G388R Variant Present	Percentage of Samples with Pathway Alteration	Percentage of Total Samples with GA and *FGFR G388R* Variant
Total	92	97	51	43%
MAPK	26	63	25	33%
NOTCH1	32	37	23	20%
HRR	39	36	24	19%
Histone/Chromatin Remodeling	24	33	24	18%
Cyclin	25	28	23	15%
PI3K	45	19	20	9%

## Data Availability

The data presented in this study are available on request from the corresponding author. The data are not publicly available due to privacy.

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
