# Peer review of "Bioinformatic Analysis of Recurrent Genomic Alterations and Corresponding Pathway Alterations in Ewing Sarcoma"

_jpm, 2023, doi:10.3390/jpm13101499_

Round 1
Reviewer 1 Report
I have carefully reviewed your paper titled "Recurrent genomic alterations and corresponding pathway alterations in Ewing Sarcoma".
The paper is well-written and provides valuable insights into this pathology, enhancing our comprehension of its complexities.
I have one specific suggestion regarding the structure of the manuscript. It appears that the abstract is quite lengthy, with an extensive introduction before delving into the study itself. While the background information is essential to providing context, it may be beneficial to streamline the abstract and introduction sections to make the paper more concise and focused. This will also help maintain the reader's interest and make the paper more accessible to a broader audience.
Aside from this structural suggestion, I find the rest of the paper to be of high quality. Your methods and results sections are clear, and the discussion provides valuable insights into the implications of your findings. I believe that once the abstract and introduction are refined, your paper will be an excellent contribution to the field.
Author Response
Thank you for the comments. We agree and have edited the abstract and introduction to make each more succinct.

Reviewer 2 Report
This is a bioinformatic analysis study on a cohort of ES patients to identify the frequency of a SNV in ES. Although, the work is of importance as this SNV has not been explored in ES before. Here are some of the concerns that should be addressed to make the study complete.
1. The title should be changed to include "Bioinformatic analysis" since the study doesn't include any validation experiments to follow up the bioinformatic analysis.
2. The method section should be expanded to include more details of bioinformatic analysis including the cut off, method of library prep and sample analysis and statistical tools used etc..
3. The references include some unintended text (ref 72-78) at the end of document should be removed.
4. The lack of validation studies using patient derived cell lines hold be mentioned as a limitation.
Author Response
1. We agree and have added this to the title of the manuscript.
2. No formal statistical software was utilized. The data was analyzed within microsoft excel. All alterations (excluding VUS) were included in the analysis and we have added reference to this in the methods section.
3. We agree and this has been removed from the manuscript.
4. We agree and this has been added as another potential limitation of the manuscript.
